# Cellular Model of Malignant Transformation of Primary Human Astrocytes Induced by Deadhesion/Readhesion Cycles

**DOI:** 10.3390/ijms23094471

**Published:** 2022-04-19

**Authors:** Roseli da S. Soares, Talita de S. Laurentino, Camila T. da Silva, Jéssica D. Gonçalves, Antonio M. Lerario, Suely K. N. Marie, Sueli M. Oba-Shinjo, Miriam G. Jasiulionis

**Affiliations:** 1Department of Pharmacology, Escola Paulista de Medicina, Universidade Federal de São Paulo (UNIFESP), São Paulo 04039-032, SP, Brazil; camilatainahdasilva@gmail.com (C.T.d.S.); jdominguesgoncalves@gmail.com (J.D.G.); mjasiulionis@gmail.com (M.G.J.); 2Cellular and Molecular Biology Laboratory (LIM15), Neurology Department, Faculdade de Medicina, Universidade de Sao Paulo, Sao Paulo 01246-903, SP, Brazil; talitalaurentino@usp.br (T.d.S.L.); sknmarie@usp.br (S.K.N.M.); 3Department of Internal Medicine, Division of Metabolism, Endocrinology and Diabetes, University of Michigan, Ann Arbor, MI 48108, USA; alerario@med.umich.edu

**Keywords:** malignant transformation, astrocytes, glioblastoma, *anoikis* resistance, tumor progression

## Abstract

Astrocytoma is the most common and aggressive tumor of the central nervous system. Genetic and environmental factors, bacterial infection, and several other factors are known to be involved in gliomagenesis, although the complete underlying molecular mechanism is not fully understood. Tumorigenesis is a multistep process involving initiation, promotion, and progression. We present a human model of malignant astrocyte transformation established by subjecting primary astrocytes from healthy adults to four sequential cycles of forced anchorage impediment (deadhesion). After limiting dilution of the surviving cells obtained after the fourth deadhesion/readhesion cycle, three clones were randomly selected, and exhibited malignant characteristics, including increased proliferation rate and capacity for colony formation, migration, and anchorage-independent growth in soft agar. Functional assay results for these clonal cells, including response to temozolomide, were comparable to U87MG—a human glioblastoma-derived cell lineage—reinforcing malignant cell transformation. RNA-Seq analysis by next-generation sequencing of the transformed clones relative to the primary astrocytes revealed upregulation of genes involved in the PI3K/AKT and Wnt/β-catenin signaling pathways, in addition to upregulation of genes related to epithelial–mesenchymal transition, and downregulation of genes related to aerobic respiration. These findings, at a molecular level, corroborate the change in cell behavior towards mesenchymal-like cell dedifferentiation. This linear progressive model of malignant human astrocyte transformation is unique in that neither genetic manipulation nor treatment with carcinogens are used, representing a promising tool for testing combined therapeutic strategies for glioblastoma patients, and furthering knowledge of astrocytoma transformation and progression.

## 1. Introduction

Astrocytes are the main class of neuroglia, performing a wide range of adaptive functions in the mammalian central nervous system (CNS). They interact with neurons, providing structural, metabolic, and trophic support [1,2]. Astrocyte reactivity is a phenomenon present in CNS pathologies, including traumatic injuries, ischemia, tumors, infectious and neuroinflammatory diseases, and neurodegenerative disorders, characterized by extensive changes in gene expression and cellular morphology [3].

Although knowledge of genetic alterations has grown in the past decade and been incorporated in the World Health Organization classification, the underlying molecular mechanism of gliomagenesis still remains to be entirely understood [4]. Of the gliomas, astrocytomas—tumors originating from astroglial lineage cells—are the most common type, representing 77.5% of all gliomas [5]. Astrocytomas are classified into four grades of malignancy, where 2–4 are defined as diffuse astrocytic tumors, and characterized by invasiveness as a deleterious aspect hampering complete surgical resection, with consequent impact on clinical outcomes. More specifically, glioblastoma (GBM)—grade 4 astrocytoma—presents rapid progression and tumor recurrence. This form has an average mortality of two years or less, in spite of the current standard of care including surgery, radiotherapy, and concomitant adjuvant therapy with chemotherapy with temozolomide (TMZ)—an alkylating cytostatic agent [6]. Concerted efforts have elucidated the major signaling pathways involved in tumor progression, and combination therapeutic strategies targeting these pathways have been employed, albeit with limited efficacy. Knowledge of the initial alterations in the malignant process may pave the way for new approaches to halt tumor progression and, ultimately, allow measures for intercepting malignant transformation and preventing perpetuation of this process.

Tumorigenesis may result from chemical, physical, or biological processes and/or genetic insults to cells—or a combination of these factors, alone or together—and can be broken down into at least three stages: initiation, promotion, and progression [7]. There is no cellular model mirroring the various stages of glioma progression in a linear fashion that allows the study of the different mechanisms and signaling pathways of the dedifferentiation of normal astrocytes into tumor cells. Previously, our group described a method of sequential deadhesion/readhesion that which resulted in malignant murine melanocyte transformation, allowing characterization of the morphological and molecular changes preceding malignant transformation. These previous findings reinforced the alteration in the microenvironment and of cell–substrate adhesion loss as a relevant factor in malignant transformation [8]. Importantly, adhesion-blocking cycles in melanocytes lead to anchorage-independent growth and epithelial–mesenchymal transition (EMT)—two steps vital to cancer progression and metastatic colonization, and also associated with *anoikis* resistance. *Anoikis*, a form of programmed cell death, occurs in anchorage-dependent cells when they detach from the surrounding extracellular matrix, and may be caused by several mechanisms, including alterations in integrins [9,10]. In order to expand our knowledge of tumorigenesis and obtain a model of malignant transformation in gliomas, primary human astrocytes were submitted to the protocol of sequential cycles of deadhesion/readhesion, as previously established for melanocytes. The resulting transformed clones exhibited unequivocal changes indicative of malignant transformation, characterized by increased proliferation and colony formation, and upregulation of enriched gene sets related to the PI3K/AKT and Wingless (Wnt)/β-catenin signaling pathways and EMT, suggesting dedifferentiation of normal astrocytes into mesenchymal-like cell types with malignant cell attributes.

## 2. Results

### 2.1. Malignant Astrocyte Transformation

Normal primary human astrocytes were checked for expression of glial fibrillary acidic protein (GFAP). Immunofluorescence analysis performed with GFAP-specific antibodies confirmed the positive expression of this astrocyte marker (Appendix A). These GFAP-positive cells were subjected to stressful conditions by sequentially blocking their adhesion to the substrate for 96 h (four sequential cycles of deadhesion and readhesion). At the end of each cycle, most cells had died from apoptosis induced by adhesion blockage (*anoikis*), while the remaining cells formed small spheroids, which were collected and cultured under normal adherent conditions. These cells generated astrocyte sublines denoted AST1c, AST2c, AST3c, and AST4c, corresponding to the sublines derived from the first, second, third, and fourth cycles, respectively. Figure 1A shows representative images of primary astrocytes together with the cell sublines derived from each deadhesion/readhesion cycle. The spheroids formed in the fourth cycle (AST4c) were submitted to a limiting dilution for clonal expansion (Figure 1B), and three different clones were randomly selected, denoted ASTcl3, ASTcl6, and ASTcl7. The short tandem repeat (STR) profiles of the AST4c cells and the cells of the three clones were similar to the profile of the original astrocytes, confirming the cell ancestry (Appendix A). Marked morphological variability was observed in the cells of the three clones (ASTcl3, ASTcl6, and ASTcl7), as shown in Figure 2A. These cells were subsequently analyzed for functional and molecular characteristics. 

### 2.2. Astrocyte-Derived Clones Acquired Tumor Cell Traits

Initially, the ability of cells to survive and proliferate from a single cell—an important characteristic of self-renewing cells—was evaluated by a colony-formation assay. All three clones (ASTcl3, ASTcl6, and ASTcl7) acquired the ability to form colonies, as evidenced after 5 days of plating (Figure 2B). ASTcl7 formed more colonies compared with ASTcl3 and ASTcl6 (Figure 2C). Notably, primary astrocytes were also analyzed, and proved unable to form colonies (data not shown).

The viability of transformed clones was evaluated under different culture conditions. The growing cell was analyzed under fetal bovine serum (FBS) starvation conditions, and with 1% and 10% FBS. ASTcl3, ASTcl6, and ASTcl7 clones cultured without FBS and with 1% FBS showed similar proliferation rates to U87MG cells in 24, 48, and 72 h. Nonetheless, differences in proliferation were evident—especially with 10% FBS—mainly for ASTcl3 and ASTcl7, which had higher proliferation rates than the U87MG cell line after 72 h in culture (Figure 3).

The ability of ASTcl3, ASTcl6, and ASTcl7 cells to form colonies in anchorage-independent growth was analyzed by a soft agar colony formation assay, widely used to evaluate cellular transformation in vitro. There was variability in the number (Figure 4A, *p* < 0.0001 for all paired-comparisons) and size (Figure 4B) of colonies formed in the anchorage-independent growth, where clone ASTcl7 formed larger colonies, but fewer in number, while ASTcl6 formed a higher number of colonies (Figure 4B). This assay confirmed the potential tumorigenic capacity of the clones. Additionally, cell migration analysis of primary astrocytes, as well as ASTcl3, ASTcl6, and ASTcl7 clones, was performed by the monolayer wound-healing assay. All three clones displayed greater migratory activity compared to the primary astrocytes after 24 h (*p* < 0.01, for ASTcl3 and ASTcl5, and *p* < 0.001, for ASTcl7). After 48 h, the scratches had completely closed for all clones (*p* < 0.0001 for all clones compared to controls), but not for the primary astrocytes (Figure 4D).

### 2.3. Astrocyte-Derived Clones’ Response to Temozolomide Treatment Proved Similar to That of U87MG Cells, with Decreased Cell Viability and Clonogenic Capacity

Given that TMZ is the drug prescribed as the GBM standard of care, the response of astrocyte-derived clones to TMZ was evaluated by 3-(4,5-dimethyl-2-thiazolyl)-2,5-di-phenyl-2H-tetrazolium bromide (MTT) and colony-formation assays. First, the 50% inhibitory concentration (IC50) of TMZ was determined for the ASTcl3, ASTcl6, and ASTcl7 clones, yielding values of 0.7223 mM, 0.7189 mM, and 0.7175 mM, respectively (Appendix A). Cell proliferation assays of the three clones were then performed with 0.72 mM TMZ for 24, 48, and 72 h (Figure 5A). A decrease in cell viability was observed for all three clones after 48 and 72 h of TMZ treatment. However, the difference was only significant for ASTcl7 relative to the respective untreated control (*p* < 0.0001 at 48 h, *p* < 0.001 at 72 h) (Figure 5A). Notably, the response of ASCTcl7 to TMZ was very similar to the response of U87MG—an established tumor cell line derived from human GBM. Additionally, the clonogenic capacity of the three clones was significantly decreased after treatment with 0.72 mM TMZ (*p* < 0.001 for ASTcl3, and *p* < 0.0001 for both ASTcl6 and ASTcl7). However, a total halt in colony formation was not achieved (Figure 5B,C).

### 2.4. Astrocyte-Derived Cell Lines Presented Activation of PI3K/AKT and Wnt Signaling Pathways, and Downregulation of Genes Related to Aerobic Respiration

High-throughput mRNA sequencing (RNA-Seq) of primary astrocytes and the three astrocyte-derived clones was analyzed for transcriptional profiling. The RNA-Seq yielded a mean of 5 million filtered reads, and detected 7651 reads presenting the ensemble gene ID. Unsupervised clustering identified six clusters with a marked difference in transcriptional profiles between the original astrocytes and astrocyte-derived clones, as demonstrated by principal component analysis (PCA) (Appendix A). The enrichment analysis of the biological processes of the Gene Ontology performed to predict possible functions of the surrogate genes in each cluster, and of the signaling pathways by KEGG analysis, revealed enrichment of a set of genes related to (1) cell proliferation/survival, (2) cell differentiation, (3) cell motility, and (4) aerobic respiration.

The upregulation of genes related to the PI3K/AKT signaling pathway was significant in the clones compared to the astrocytes, starting with the upregulation of genes coding for ligands such as insulin growth factor 1 (*IGF1*) and platelet-derived growth factor subunit B (*PDGFB*), and the respective receptors *IGF1R* and platelet-derived growth factor receptor alpha (*PDGFRA*). Genes coding for components of the complex formed by receptor tyrosine kinases—fibroblast growth factor receptor substrate 2 (FRS2), growth-factor receptor-bound protein-2 (*GRB2*), and GRB2-associated binding protein 1 (*GAB1*)—were also upregulated. Additionally, genes of the downstream targets—such as phosphatidylinositol-4-5-biphosphate 3-kinase catalytic subunit alpha, phosphoinositide-3-kinase regulatory subunit 1 (*PI3KR1*), 3-phosphoinositide-dependent protein kinase 1 (*PDPK1*), AKT serine/threonine kinase 3 (*AKT3*), mechanistic target of rapamycin kinase (*MTOR*), and RPTOR-independent companion of MTOR complex 2 (*RICTOR*)—were upregulated, as were genes coding for the transcription factor and cAMP-responsive element-binding protein 1 (*CREB1*). The upregulation of these genes convergently leads to cell growth and survival. Interestingly, forkhead box O3 (*FOXO3*), which codes for a transcription factor that triggers for apoptosis, was downregulated in the clones compared to astrocytes, thereby contributing to clone cell survival. By contrast, genes related to other signaling pathways regulating cell proliferation—such as transforming growth factor beta (*TGFB1*), corresponding receptors (*TGFBR1* and *2*), and epidermal growth factor receptor (*EGFR*)—were all downregulated.

Notably, the gene components of the Wingless/integrated (Wnt) canonical pathway were extensively upregulated, from external cell components—such as the ligand and receptors—to nuclear signaling: the secreted signaling protein Wnt family member 4 (*WNT4*), LDL receptor-related proteins 5 and 6 (*LRP5* and *LRP6*), frizzled class receptors 1 and 4 (*FZD1* and *FZD4*), axin 1 (*AXIN1*), APC regulator of the Wnt signaling pathway (*APC*), glycogen synthase kinase 3 beta (*GSK3B*), β-catenin (*CTNNB1*), and the transcription factors activated by nuclear β-catenin—*TCF7L1* and *TCF7L2*—and lymphoid enhancer-binding factor 1 (*LEF1*). 

Moreover, genes that code for other transcription factors involved in the regulation of embryonic development and in the determination of cell fate were upregulated in the clones, including SRY-box transcription factors 2 and 4 (*SOX2* and *SOX4*) and Krüppel-like factor 4 (*KLF4*), involved in the regulation of differentiation and somatic cell reprogramming. Interestingly, the genes that induce the EMT were also upregulated, i.e., the transcription factors zinc finger E-box binding homeobox 1 and 2 (*ZEB1* and *ZEB2*), and Twist family bHLH transcription factor 1 (*TWIST1*). Additionally, genes coding for mesenchymal markers such as N-cadherin (*CDH2*), vimentin (*VIM*), and fibronectin (*FN1*) were upregulated, although not uniformly in the three clones.

In Figure 6A,B, upregulation of the PI3K/AKT and Wnt/β-catenin signaling pathways, as well as the consequent EMT and stemness acquisition of astrocyte-derived clones, are presented as a heatmap and a schematic diagram with all differentially expressed genes.

In contrast, the downregulation of genes related to mitochondrial oxidative metabolism was significant in the clones compared to astrocytes, as denoted by the enrichment of genes related to generation of precursor metabolites and energy (GO 6091, adjP = 6.47× 10^−7^), aerobic respiration (GO 9060, adjP = 1.04 × 10^−5^), the electron transport chain (GO 22900, adjP-1.1 × 10^−5^), and cellular respiration (GO 45333, adjP = 5.65 × 10^−5^) (Figure 7A). Particularly, 23 genes coding for subunits of complex I of oxidative phosphorylation (OXPHOS), 4 genes coding for succinate–ubiquinone oxidoreductase (SDH) of complex II, 6 genes coding for ubiquinol–cytochrome c reductase of complex III, 8 genes coding for cytochrome c oxidase of complex IV, and 8 genes coding for ATP synthase of complex V were downregulated in the clones (Figure 7B).

### 2.5. Astrocyte-Derived Cell Lines Exhibited Increased GFAP and PDGFRA Protein Expression

The expression of the GFAP protein was increased in the astrocytes submitted to deadhesion/readhesion cycles, with greater intensity in cells after the first cycle (Appendix A). GFAP expression was also higher in the cells of the three clones relative to the primary astrocytes (Figure 8A,C). Similarly, the expression of PDGFRA was significantly higher in ASTc1, ASTc2, and ASTc4 (Appendix A), as well as in ASTcl3 and ASTcl7, compared to primary astrocytes, validating the gene expression findings (Figure 8B,C).

## 3. Discussion

Tumorigenesis is a multistep process that includes invasive behavior, deregulated growth, morphological and histological changes, and cumulative burden of a diversity of somatic mutations [11]. This process can be broken down into at least three stages: an initiation step involving irreversible genetic alterations that result in dysregulation of signaling pathways, a promotion step with cell proliferation through promoter–receptor interaction, and a progression step characterized by malignant transformation and growth [7]. Several models of tumorigenesis have been developed aiming at expanding knowledge of neoplastic progression, e.g., 3D co-culture models [12,13], 3D bioprinters [14], a model based on live-cell-generated oxygen and nutrient gradients [15], and a model based on *anoikis* resistance [8]. Thus far, however, none of these cellular models has yet been applied to study glioma progression. Astrocytoma is the most common glioma, and GBM is its most aggressive form, with a poor prognosis despite extensive research effort over the last decade [5]. In this context, a model that allows the study of astrocytoma tumorigenesis may help us to gain insights into innovative therapeutic strategies. 

Cancer cells rapidly develop several mechanisms to circumvent obstacles impinging their survival, including development of resistance to *anoikis* [16]. In a previous study, we developed an experimental methodology of sequential cycles of forced anchorage impediment, which induced malignant transformation in murine melanocytes [8]. In the present study, the same approach was applied to primary human astrocytes, blocking their adhesion to the substrate in sequential cycles. Mirroring previous results, astrocytes submitted to four adhesion-blockage cycles presented significant morphological and behavioral changes. These astrocyte-derived stressed cells formed more colonies that were larger in size compared to the original astrocytes, characterized by clonal expansion—an important feature of tumorigenesis [17,18]. One of the hallmarks of cancer is the ability to induce and positively sustain growth-stimulatory signals. Cancer cells must also circumvent powerful programs that negatively regulate cell proliferation [19,20]. The clones resulting from adhesion blockage exhibited a high proliferative capacity under different nutritional conditions—even starvation—and demonstrated similar proliferative capacity to U87MG cells—a GBM cell line. The PI3K/AKT/mTOR pathway is one of the most relevant cascades activated in GBM related to tumor growth, metabolism, cell proliferation/survival, angiogenesis, autophagy, and chemotherapy resistance [21]. Several transcripts of growth factors (*IGF1*, *PDGFB*) and respective tyrosine kinase receptors (*IGF1R*, *PDGFRA*) and docking proteins (*FRS2*, *GRB2,* and *GAB1*)—active initiators of the PI3K/AKT pathway—were found to be upregulated in the transformed clones in the RNA-Seq analysis. In fact, *PIK3CA* encoding the p110a subunit of PI3K, and *PIK3R1* encoding p85 regulatory subunit of PI3K, as well as *AKT3*, were upregulated. These stimulate cell cycle progression, inhibit apoptosis and, subsequently, promote cell growth through activation of mTOR signaling [22]. PI3K/AKT and their mammalian target of mTOR are key regulators of metabolic reprogramming towards glycolysis, stabilizing HIF-1α which, in turn, increases the transcription of some key glycolytic enzymes [23]. Aerobic glycolysis (the Warburg effect) is one of the most common forms of metabolic reprogramming of cancer cells, including GBM cells, as opposed to oxidative phosphorylation in normal cells [24]. This shift provides carbon and nitrogen sources for amino acids, fatty acids, and other metabolites necessary for cancer cell survival and proliferation—even under unfavorable conditions [25]. A recent study has shown that this glycolytic shift promoting GBM progression occurs in a PDPK1-dependent manner [26]. Interestingly, *PDPK1* was also upregulated in the clones analyzed in the present study. CREB, activated by AKT, is a vital regulator of cyclin D1 expression in GBM cells, and PI3K-CREB signals have been implicated as important for regulating the invasive behavior of GBM cells [27]. FOXO3A, another host target of AKT, may act synergistically with HIF-1α and induce apoptosis [28]. Thus, *FOXO3A* downregulation in the clones may contribute to enhancing their survival.

Moreover, the canonical Wnt signaling pathway, which regulates the stemness of normal cells, was extensively upregulated in the transformed clones. The canonical Wnt components, after binding to cell surface receptors (FZD), activate the key complex comprising APC, GSK3β, and axin, and stabilize β-catenin, which translocates to the nucleus and, by forming a compound with the TCF/LEF family of transcription factors, activates Wnt target genes, including cyclin-D, c-Myc, and VEGF. The emergence of cancer stem cells with enhanced proliferation has been associated with alterations in the Wnt signaling pathway [29]. Additionally, β-catenin regulates cell–cell adhesion by binding to cadherin—a component of the cell adhesion complex. An increase in this binding promotes glioma cell migration and EMT. EMT was first described during embryonic development [30], and is associated with extensive changes in the expression of genes regulating cell adhesion, the cytoskeleton, the extracellular matrix, cellular junctions, and signaling pathways [31]. As a consequence of these alterations, polarized cells acquire a mesenchymal phenotype characterized by anchorage-independent growth and migratory properties—shifts that were observed in the three transformed clones. Although these clones formed colonies, differences in the size and number of colonies were evident, suggesting possible genetic and/or epigenetic heterogeneity. Additionally, all transformed clones demonstrated a high rate of cell migration in comparison to the primary astrocytes. Thus, *anoikis* resistance is a characteristic EMT process. The Wnt signaling pathway also activates TWIST1—a member of the basic helix–loop–helix transcription factor family involved in cell differentiation and lineage determination, but that also facilitates cancer metastasis by acting as an EMT-inducing transcription factor [32]. TWIST1 is negatively regulated by FOXO3A [33]. *TWIST1* was upregulated in two of our clones, and was inversely correlated with *FOXO3A* expression. Additionally, in our clones, *ZEB1* and *ZEB2*—transcription factors that attach to the E-cadherin promoter region and inhibit the expression of this adhesion molecule—were upregulated, thereby increasing tumor cell motility. Higher recurrence was observed among GBMs that had high *ZEB2* levels [34], while *ZEB2* also directed EMT-like processes in glial responses to injury [35], including alternative cell fate decisions, proliferation, and differentiation [36]. Similarly, glioma cells with *ZEB1* expression induced by nuclear factor-β were more invasive [37]. 

SOX genes encode transcription factors, DNA-binding proteins involved in sex determination, growth, proliferation, survival, and stem cell pluripotency [38]. SOX2 is a member of the SOXB1 subcategory, which promotes the maintenance of embryonic stem cells, interacting with OCT4, KFL4, c-Myc and Nanog [39]. In addition, SOX2 regulates cyclin D1 and CD133 in cancer cells, affecting cancer cell proliferation and generation [40]. In the present study, *SOX2* was upregulated in two of the clones, and both *SOX4* and *KFL4* expression were upregulated in all three clones, suggesting involvement of these targets in the differentiation process of the transformed clones. *FN1* is highly expressed in GBM and linked to the extracellular matrix ultrastructure [41,42]. FN1 leads to activation of the PI3K/AKT signaling pathway in an integrin-dependent manner, resulting in proliferation, survival, invasion, and chemoresistance [43]. Moreover, a more differentiated phenotype is observed in glioma stem-like cells exposed to FN1, with decreased levels of SOX2 and increased levels of GFAP [44], highlighting the importance of FN1 for glioma onset and progression. Interestingly, *FN1* was upregulated in all three transformed clones.

Metabolic reprogramming was observed in the clones with downregulation of genes involved in the mitochondrial oxidative phosphorylation, particularly with the genes related to electron transport in comparison to normal astrocytes. The consequent decoupling of the electron transport led to reactive oxygen species production, which can leverage the malignant transformation. The undifferentiated stemness state acquired by the astrocyte-derived clones could explain this metabolic reprogramming as a protective state, as impaired mitochondria induce stem cell senescence [45,46].

The presence of GFAP—a structural protein of the glial cytoskeleton and a mature astrocyte marker [47,48]—was detected in the transformed clones, confirming their origin from primary human astrocytes. GFAP is present in GBM stem-like cells [49], and its expression has been correlated with a poorer prognosis [50].

The transformed clones were susceptible to TMZ treatment—a standard chemotherapy for GBM treatment [6]—showing a similar response pattern to that of U87MG cells. TMZ decreased colony size and numbers in the clones analyzed, but failed to kill the transformed cells of the clones, suggesting resistance to treatment. The activation of autophagy by TMZ has been described as contributing to GBM therapeutic resistance, and combination therapeutic strategies using TMZ and small-molecule inhibitors of the Wnt/β-catenin pathway have been proven to enhance efficacy in GBM cells [51]. The present methodology for obtaining astrocyte-derived cells that present traits of malignancy, with activation of both the PI3K/AKT and Wnt/β-catenin pathways, may help to test further combination therapeutic strategies targeting pivotal signaling pathways in the search for synergistic effects in GBM.

Further studies aimed at identifying epigenetic mechanisms underlying the molecular and behavioral changes in clonal cells relative to original astrocytes are warranted, in an effort to widen the spectrum of “druggable targets” for GBM.

## 4. Materials and Methods

### 4.1. Cell Lines and Cell Culture

A commercial human GBM strain, U87MG, was obtained from the American Type Culture Collection (ATCC) and grown in monolayer in Dulbecco’s Modified Eagle’s Medium (DMEM) (Thermo Fisher Scientific, Carlsbad, CA, USA) supplemented with 10% inactivated fetal bovine serum (FBS) (Cultilab, Campinas, Brazil) and antibiotic (100 units/mL penicillin, 100 µg/mL streptomycin). Primary human astrocytes (K1884) were purchased (Gibco by Life Technologies, Madison, MI, USA) and maintained in a specific medium for astrocyte growth (DMEM 1×, GlutaMAX, N-2 Supplement, One Shot FBS) (Thermo Fisher Scientific) at 37 °C, in an atmosphere containing 5% CO_2_. Cell sublines obtained after subjecting primary astrocytes to 1, 2, 3, and 4 deadhesion/readhesion cycles (AST1c, AST2c, AST3c, and AST4c, respectively), as well as clones (ASTcl3, ASTcl6, and ASTcl7) originating after limiting dilution of astrocytes subjected to four deadhesion/readhesion cycles, were cultured in DMEM with 10% FBS and antibiotics. Cells were washed with phosphate-buffered saline [52] and removed from the plate via the addition of a 0.2% trypsin solution (Thermo Fisher Scientific) after reaching about 70% confluence.

### 4.2. Immunofluorescence Analysis

Cells were plated on coverslips previously covered with poly-L-lysine (Sigma-Aldrich, St. Louis, MO, USA) and fixed for 1 h and 30 min with 4% paraformaldehyde in 1× PBS at 4 °C. Cells were washed with PBS three times and then permeabilized with 0.1% NP-40 diluted in PBS for 30 min at 37 °C and washed twice with PBS. Nonspecific sites were blocked by incubation with 4% goat serum (Sigma-Aldrich) for 30 min at 37 °C, followed by 2 washes with PBS. Cells were incubated with primary anti-GFAP antibody (1:100, Thermo Fisher Scientific) and anti-PDGFRA (1:200, Abcam), followed by Alexa Fluor 488-labeled secondary antibody or Alexa Fluor 568-labeled secondary antibody (1:100, Thermo Fisher Scientific), diluted in blocking solution in a humid chamber for 48 h at 4 °C. After incubation, cells were washed 3 times with PBS. The nuclei were labeled with DAPI 300 nM (diamidino-2-phenyl-indole, Thermo Fisher Scientific) for 3 min, followed by three washes with PBS. Negative controls included the complete reaction with no primary antibody. Documentation and analysis of the slides were performed by fluorescence microscopy using a 40× objective (2× zoom). Fluorescence quantification was performed by integrated density via the selection of regions of interest (ROIs) using ImageJ software (National Institutes of Health, Bethesda, MD, USA). Measurement of the total area was performed by selection in the regions of interest using bright-field microscopy. For each cell condition, 60–100 ROIs/cell were used. The following equation of corrected total cell fluorescence (CTCF) was used: CTCF = integrated density (ROI) − (area of ROI × mean fluorescence of background readings) [53].

### 4.3. Cell Adhesion Blockade (Anoikis Resistance)

Human primary astrocytes (1 × 10^5^ cells/mL) were grown on plates coated with 1% sterile agarose to prevent cell–extracellular matrix adhesion in a specific medium for astrocyte growth (Sigma-Aldrich) for 96 h at 37 °C, in an atmosphere containing 5% CO_2_. Surviving non-adhered cells formed spheroids (resistant to *anoikis*), which were collected and plated in standard culture conditions favoring adhesion (DMEM, with 10% FBS). The cell lines AST1c, AST2c, AST3c, and AST4c were obtained after subjecting normal astrocytes to 1, 2, 3, and 4 deadhesion/readhesion cycles, respectively. After the fifth adhesion–impediment cycle, the obtained spheroids were plated in 96-well plates by limiting dilution, so that each well contained only 0.5–1 spheroids. At least three clones (i.e., ASTcl3, ASTcl6, and ASTcl7) were randomly selected for expansion and functional and molecular analyses.

### 4.4. DNA and RNA Extraction

For extraction of genomic DNA and total RNA, cells were lysed in AllPrep DNA/RNA Mini Kit lysis buffer, following the protocol of the manufacturer (Qiagen, Valencia, CA, USA). The quality of extracted DNA and RNA, along with their concentrations and purity, was determined by reading the absorbance at 260 and 280 nm with a spectrophotometer (NanoDrop, Thermo Fisher Scientific). A260/A280 ratios greater than 1.8 were considered satisfactory for purity. DNA was diluted and stored for use in cell authentication. The RNA samples were stored at −80 °C until their use for the synthesis of complementary DNA (cDNA).

### 4.5. Complementary DNA Synthesis by Reverse Transcription

Reverse transcription was initially performed with DNase digestion to eliminate any contamination with genomic DNA. Transcription was performed with SuperScript III reverse transcriptase, RNase inhibitor (RNaseOUT), random oligonucleotides, and oligo(dT), following the recommendations of the manufacturer (Thermo Fisher Scientific). The cDNA obtained was treated with 1 U of RNase H at 37 °C for 30 min and 72 °C for 10 min to eliminate any hybrids formed. cDNA was then diluted in Tris-EDTA buffer and stored at −20 °C for subsequent analysis of genes expressed in these cells.

### 4.6. DNA and RNA Extraction

The genetic verification of cells was performed by analysis of STR regions of the genomic DNA, using the GenePrint 10 System kit (Promega, Madison, WI, USA). The DNA was quantified and diluted. For the pre-amplification reaction, 5 µL of Primer Pair Mix, 5 µL of Master Mix, and 10 ng of DNA in 15 µL were added to the microtube. Samples were submitted to the temperature cycles recommended by the manufacturer. PCR products were analyzed on the 3500 Genetic Analyzer (Thermo Fisher Scientific), and the sequenced fragments (containing the 9 STR loci and amelogenin) were analyzed using GeneMapper software version 4.1 (Thermo Fischer Scientific), following the recommendations of the manufacturer. Similarly, U87MG was analyzed for authenticity, and the results based on similarities were checked against the database provided by the ATCC.

### 4.7. Acquired TMZ Sensitivity

The U87MG cell line and ASTcl3, ASTcl6, and ASTcl7 clones were treated with different TMZ concentrations (0.4, 0.8, 1.6, 3.2, or 6.4 mM) for up to 72 h. The concentration used in further experiments was based on the IC50 (50% inhibitory concentration) of TMZ. Cell viability was determined after 48 h of TMZ treatment using the IC50.

### 4.8. Colony Formation Assay

In total, 50 cells of each clone were plated on 6 cm diameter plates under normal culture conditions. We performed two independent experiments in triplicate. After 10 days, cells were washed with PBS and fixed with 3.7% formaldehyde in PBS for 15 min. Colonies were stained with 1% crystal violet. The plates were washed three times with distilled water and photographed. For colorimetric quantification, 1 mL of 1% SDS was added for 4 h, and absorbance reading was recorded at 620 nm.

The sensitivity of clones to TMZ treatment was analyzed by plating 100 cells/well in 6-well plates. At day five, cells were treated with TMZ for 48 h. Colonies were diluted and plated in a 96-well plate, and 20 µL of 5 mg/mL MTT was added and incubated at 37 °C with 5% CO_2_. After 3 h, the solution was removed, and 100 µL of dimethyl sulfoxide was added. The plate was shaken for 30 min at room temperature, and the absorbance was read at 620 nm with an ELx808 microplate reader (BioTek, Winooski, VT, USA). Experiments were performed in triplicate in two independent assays.

### 4.9. Cell Viability

Cell viability was determined daily for 3 consecutive days after plating 2.0 × 10^4^ cells per well in a 96-well plate, using the PrestoBlue Cell Viability Reagent (Thermo Fisher Scientific). ASTcl3, ASTcl6, and ASTcl7 clones were analyzed with and without FBS (1% and 10%) stimulation. Additionally, the viability of clones was also compared in relation to TMZ treatment response. U87MG cells were used as a reference glioma cell line. Fluorescence intensity (excitation at 540 nm, emission at 560 nm) was measured using a GloMax-96 Microplate Luminometer (Promega). The background consisting of cell culture medium was measured for each plate and subtracted from each measurement value. The viability change was calculated as the fold change in relation to the control at time zero. Experiments were performed in octuplicate in two independent analyses.

### 4.10. Wound-Healing Assay

In total, 2 × 10^5^ cells/well were cultured in 24-well plates until they reached monolayer confluence. The culture medium was removed and a discontinuity was made in the monolayer using a 200 µL micropipette tip, creating a cell-free area (“wound”). The wells were washed with PBS to remove the loose cells, and then culture medium with 1% FBS was added to prevent cell proliferation. The closure of the scratches was recorded at different times (0, 8, 24, and 48 h), in triplicates of two independent experiments. Reference points were marked on the bottom of each well of the plate to locate the fields corresponding to the cell-free areas analyzed for imaging at the different time intervals. Cell-free areas were first calculated as a percentage of the cell-free area at time 0, and arbitrarily marked as 100%. Then, the percentage of free area was calculated using ImageJ. The percentage of invaded area at the different times was determined by the difference in the area at time zero (100%) minus the free area [54].

### 4.11. High-Throughput Sequencing for Transcriptome Analysis

The transcriptomic profiles of primary astrocytes and the three clones were analyzed by RNA-Seq in quadruplicate. Libraries were prepared using the QuantSeq 3′mRNA-Seq Library Prep Kit-FWD by Illumina (Lexogen, Vienna, Austria), using 500 ng of tumor RNA, following the recommendations of the manufacturer. The library concentration was measured using the Qubit dsDNA HS Assay Kit (Thermo Fisher Scientific), and the size distribution was determined using the Agilent D1000 ScreenTape System on TapeStation 4200 (Agilent Technologies, Carlsbad, CA, USA). Sequencing was performed on the NextSeq 500 platform at the next-generation sequencing facility core SELA at the School of Medicine of the University of São Paulo. Sequencing data were aligned to the GRCh38 version of the human genome with STAR [55]. Downstream processing of the BAM files (merging of different lanes; marking of duplicates) was performed with the bammarkduplicates tool from biobambam2; featureCounts was employed to count the number of reads that overlapped each gene [56]. The GFF file containing the gene models was obtained from ftp.ensembl.org. Sequencing quality and alignment metrics were assessed with FastQC and RNASEQC, respectively. Downstream analyses were performed in R using specific Bioconductor and CRAN tools, and briefly described. Normalization was performed with *edgeR* using the trimmed-mean (TMM) method. We used sva to remove occult/unwanted sources of variation from the data. The R-Bioconductor package *limma* was used to assess differential gene expression in each group, and to perform log2 counts per million reads mapped (CPM) in the transformation of the data. Principal component analysis was performed using the prcomp function from R-stats, and graphically depicted as biplots constructed using ggplot2. To identify modules of co-regulated genes among the differentially expressed genes, we used pheatmap and cutree to perform hierarchical clustering and to build heatmaps displaying these modules. We used Manhattan distance as the similarity metrics, and the ward D2 clustering algorithm. We used clusterProfiler to perform gene set enrichment analysis for each module of co-regulated genes. Expression data were centered to the mean of each gene.

### 4.12. Statistical Analyses

Statistical analyses were performed using GraphPad Prism software, version 8. For the analyses of gene expression, proliferation, colony formation, monolayer wound assay, and soft agar colony formation, the two-way ANOVA test was used for comparison of multiple groups, followed by Tukey’s post hoc test. For IC50 results, the nonlinear parameter (regression inhibitor vs. normalized response) was used. Values of *p* < 0.05 were considered significant.

## 5. Conclusions

The sequential deadhesion/readhesion of primary human astrocytes resulted in the clonal expansion of astrocyte-derived GFAP-positive cells with malignant traits, including high proliferative and migratory rates—comparable to a human GBM cell line (U87MG)—in addition to clonogenic capability and anchorage-independent growth. These transformed cells presented activation of the PI3K/AKT and Wnt/β-catenin signaling pathways, leading to cell proliferation, survival, apoptosis avoidance, and a mesenchymal-like phenotype, with upregulation of genes related to EMT. They also presented downregulation of oxidative phosphorylation, indicative of metabolic reprogramming. The functional and molecular characteristics of these transformed clonal cells derived from primary human astrocytes may be useful for modeling combinational therapeutic strategies to improve the outcomes of GBM patients.

## Figures and Tables

**Figure 1 ijms-23-04471-f001:**
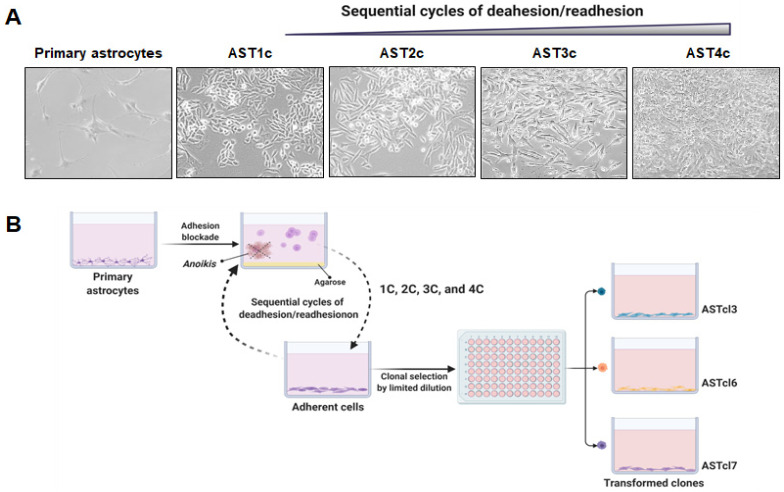
Astrocyte malignant transformation induced by cycles of deadhesion/readhesion: (**A**) Representative images of primary astrocytes and astrocytes submitted to one (AST1c), two (AST2c), three (AST3c), and four (AST4c) sequential deadhesion/readhesion cycles (magnification of 40×). (**B**) Experimental approach to establish the cellular model of malignant transformation of primary human astrocytes. Primary astrocytes were subjected to sequential cycles of deadhesion/readhesion, and after four cycles the spheroids obtained were plated by limiting dilution in 96-well plates. Different clones were randomly selected, giving rise to three distinct clones, denoted ASTcl3, ASTcl6, and ASTcl7. Figure created with BioRender.

**Figure 2 ijms-23-04471-f002:**
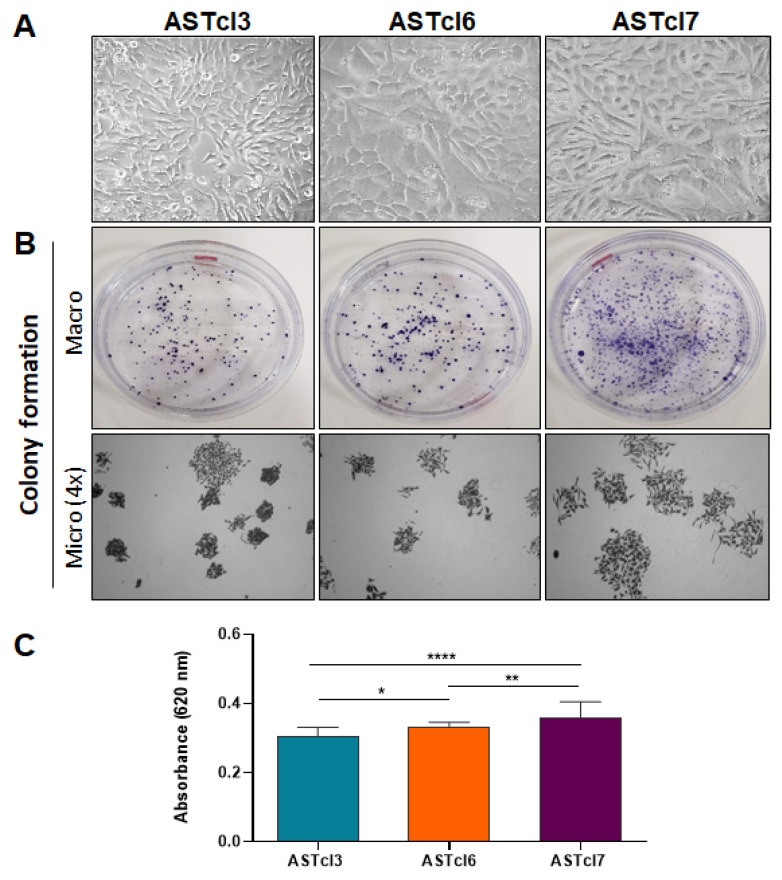
Clones obtained after subjecting primary astrocytes to sequential cycles of deadhesion/readhesion presented altered cellular morphology and colony-formation capability: (**A**) Photomicrographs of cell cultures of ASTcl3 (clone 3), ASTcl6 (clone 6), and ASTcl7 (clone 7). (**B**) Colony formation after 10 days, stained with crystal violet, and shown in a micro view and at a magnification of 4×. (**C**) Graph representing the mean ± standard deviation of four replicates of two independent colony-formation experiments after colorimetric measures of stained colonies. The asterisks indicate statistically significant differences: * *p* < 0.5, ** *p* < 0.01, **** *p* < 0.0001.

**Figure 3 ijms-23-04471-f003:**
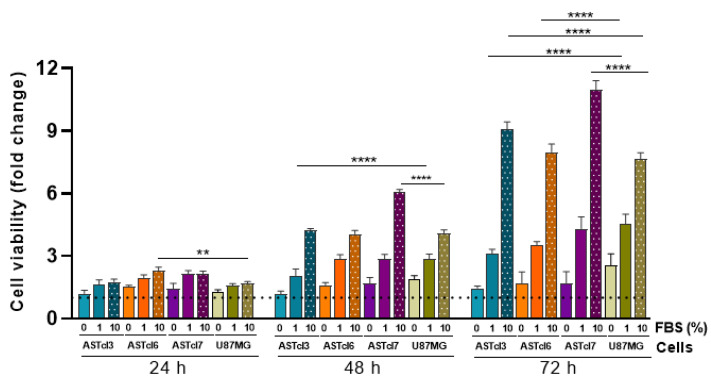
Astrocyte-derived clones acquired the capability to survive under serum starvation, similar to U87MG GBM cells. Graph showing cell viability in the absence or presence of fetal bovine serum (FBS) (1% and 10%) at 24, 48, and 72 h. Bars represent the means ± standard deviations of the fold change relative to the control at time zero. Statistically significant differences between the clones (ASTcl3, ASTcl6, and ASTcl7) in each condition (0, 1 and 10% FBS), compared to U87MG GBM cells, are represented by asterisks: ** *p* = 0.01, **** *p* < 0.0001.

**Figure 4 ijms-23-04471-f004:**
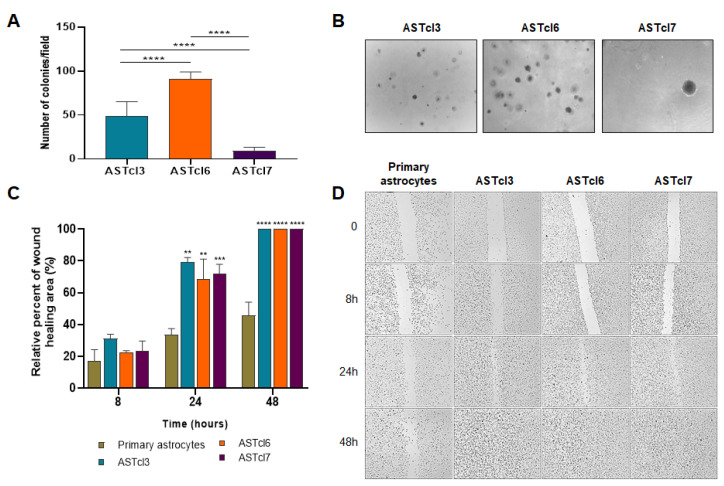
ASTcl3, ASTcl6, and ASTcl7 cells were able to grow independently of anchorage and migrate collectively. The capability of astrocyte-derived clones (ASTcl3, ASTcl6, and ASTcl7) to form colonies in soft agar is shown in (**A**) (number of colonies) and (**B**) (colonies’ size; magnification of 4×). The bars represent the average number of colonies per well ± standard deviation of two independent experiments, in triplicate. (**C**) Graph representing the area occupied after migration by ASTcl3, ASTcl6, and ASTcl7 cells compared to primary astrocytes at 0, 8, 24, and 48 h after scratching. (**D**) Representative photomicrographs (10× magnification) of wound healing of primary astrocytes and ASTcl3, ASTcl6, and ASTcl7 cells at 0, 8, 24, and 48 h. The results are expressed as the means ± standard deviations (triplicate) of two independent experiments. Asterisks indicate statistically significant differences between ASTcl3, ASTcl6, and ASTcl7 in relation to primary astrocytes at different times: ** *p* < 0.001, *** *p* < 0.0001, and **** *p* < 0.00001.

**Figure 5 ijms-23-04471-f005:**
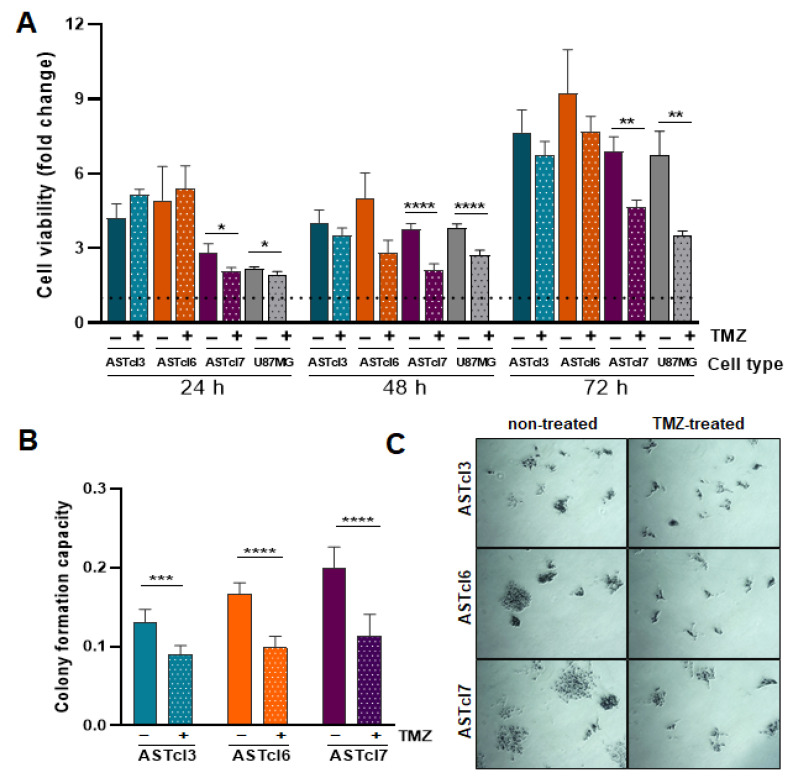
Response of ASTcl3, ASTcl6, and ASTcl7 clones to TMZ treatment: (**A**) Cell viability of ASTcl3, ASTcl6, ASTcl7, and U87MG in the absence (−) and presence (+) of temozolomide (TMZ). Bars represent the means ± standard deviations of the fold change relative to the untreated cells at time zero. (**B**) Colony-formation assay of ASTcl3, ASTcl6, and ASTcl7 clones treated after 5 days with 0.72 mM TMZ per 48 h. Bars represent two independent experiments performed in triplicate, and are expressed as the mean ± standard deviation. (**C**) Representative photomicrographs of colony-formation assays. Statistically significant differences are represented by asterisks: * *p* < 0.05, ** *p* < 0.01, *** *p* < 0.001, and **** *p* < 0.0001.

**Figure 6 ijms-23-04471-f006:**
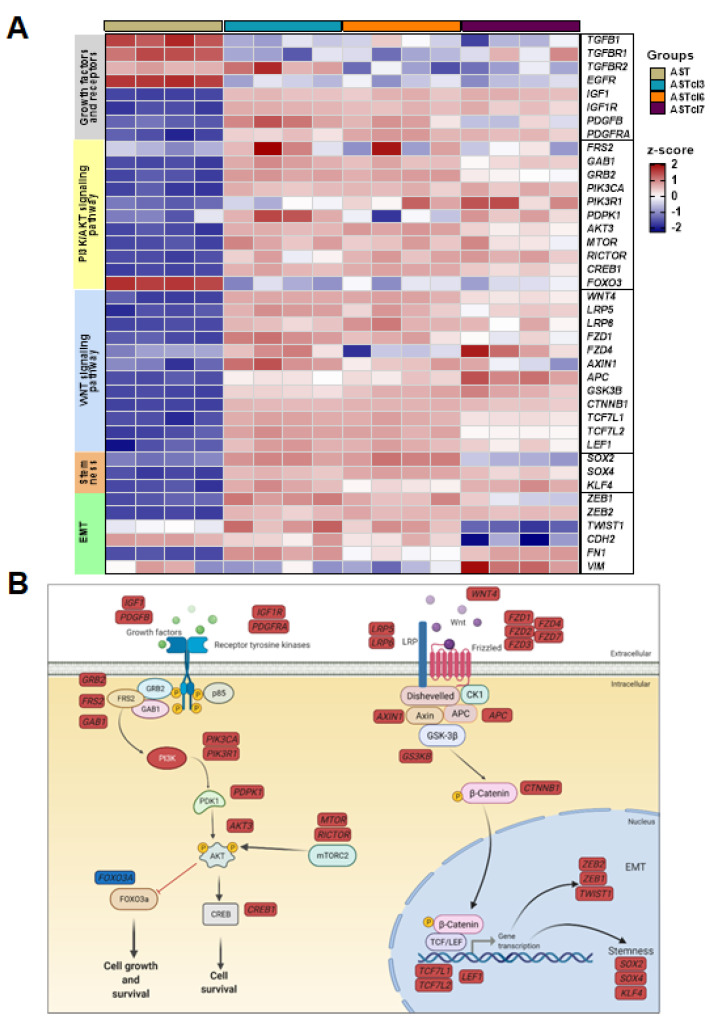
Transcriptome analysis of astrocyte-derived cells: (**A**) Heatmap representing the expression of growth factors and receptors, PI3K/AKT and Wnt signaling pathways, and stemness and EMT genes from RNA-Seq data of primary astrocytes (AST) and derived clones (ASTcl3, ASTcl6, and ASTcl7). (**B**) The PI3K/AKT and Wnt/β-catenin signaling pathways and the differentially expressed genes (upregulated and downregulated in clones in relation to astrocytes in red and blue, respectively) represented in the heatmap. Activation of these pathways contributed to cell growth and survival and epithelial–mesenchymal transition (EMT), along with the stemness state of astrocyte-derived clones by *anoikis* resistance. Figure created with BioRender.

**Figure 7 ijms-23-04471-f007:**
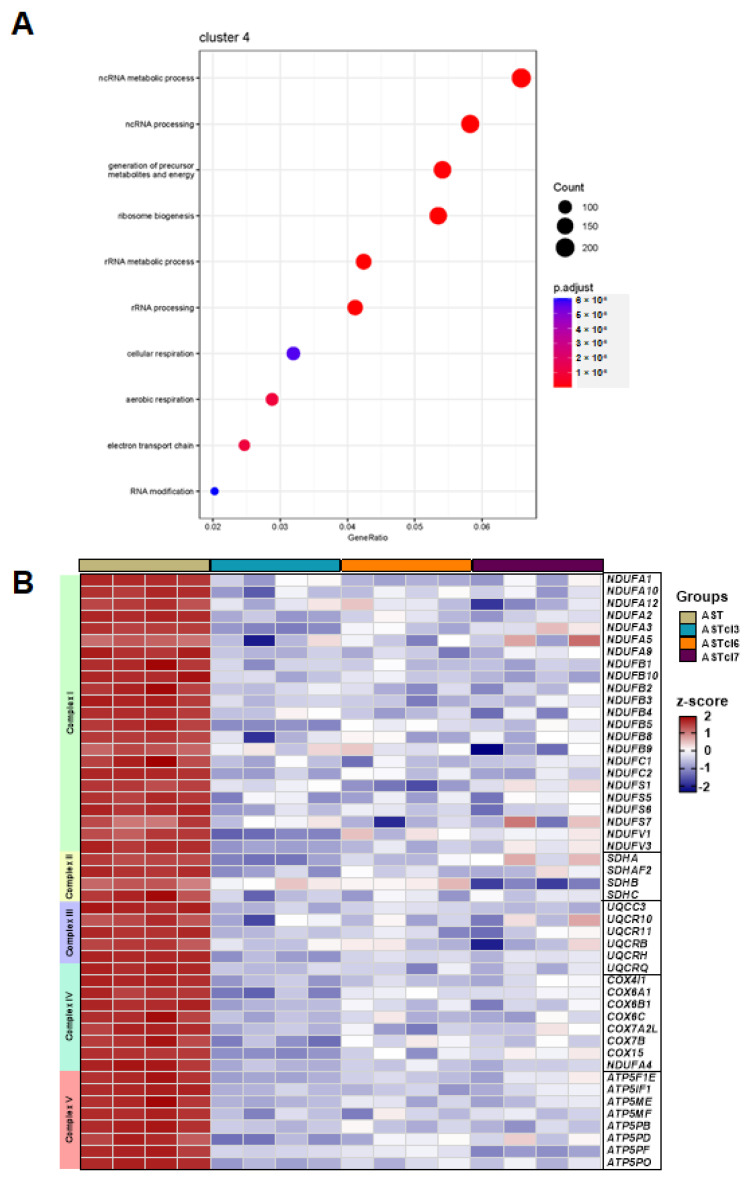
Transcriptome analysis of astrocyte-derived cells: (**A**) Dot plot of the 10 most enriched GO processes of cluster 4 (Appendix A). (**B**) Heatmap representing the expression of genes coding for components of subunits of the five complexes of oxidative phosphorylation presented in the GO: generation of precursor metabolites and energy (GO 6091), cellular respiration (GO 45333), aerobic respiration (GO 9060), and the electron transport chain (GO 22900).

**Figure 8 ijms-23-04471-f008:**
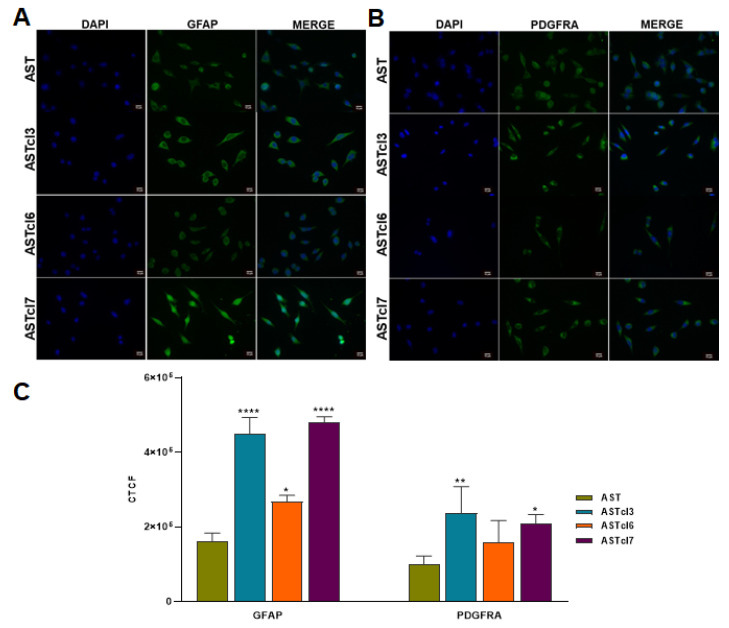
Immunofluorescence analysis in primary astrocytes and derived clones: (**A**) Immunofluorescence staining for GFAP (green) and nuclei (DAPI; blue). (**B**) Immunofluorescence staining for PDGFRA (green) and nuclei (DAPI; blue). (**C**) Quantification of GFAP and PDGFRA staining in primary astrocytes and clones derived from deadhesion blocking cycles (ASTcl3, ASTcl6, and ASTcl7). CTCF: corrected total cell fluorescence. Magnification of 40×, scale bar = 10 µm. Statistically significant differences in each clone compared to primary astrocytes are represented by asterisks: * *p* < 0.05, ** *p* < 0.01, and **** *p* < 0.0001.

## Data Availability

Not applicable.

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
