# Peer review of "Cellular Model of Malignant Transformation of Primary Human Astrocytes Induced by Deadhesion/Readhesion Cycles"

_ijms, 2022, doi:10.3390/ijms23094471_

Round 1

Reviewer 1 Report

The manuscript submitted by primary author Soares et al highlights the development and characterization process of a cellular model of transformed GBM/glioma from a parental population of primary human astrocytes. At large this publication can be very useful in understanding the complex process of malignant transformation and can be used as a tool to study the progression of pathogenesis features, as well as developing therapeutics. However, the authors have left unaddressed a significant deal of details. To define these cells useful for such applications and also characterize as a model of GMB/glioma the authors should meticulously address the following:

1.       Throughout the paper authors have NOT included appropriate control cells, the parental primary astrocytes (except Fig 4 and 6). To characterize the feature described in Fig1, fig3, fig5, proper control has be included. Any phenotype they describe in the manuscript should to be presented in relativity to the primary cells.

2.       Since the design of the study is focused on stressing the cells with cycles of deadhesion/readhesion, have the authors looked into whether the cells show hallmarks of EMT and whether the cells show aberrant expression profile compared to primary astrocytes?

3.       Line 174: please define clonogenic capacity, it’s the first time authors introduced this term. This term can be used in an abstract manner, therefore please explain the protocol of quantification.

4.       Fig5: The authors describe a significant decrease in cell number upon TMZ treatment. The data represents sensitive cells within the selected clones. However, in the discussion (line 238) they describe the clones to be TMZ resistant. For such conclusions the authors should extend the experiment, they can do a longer period of treatment and record the changes in TMZ resistance.

5.       If the authors can include some kind of qualitative or quantitive methods to show the expression levels of Oct4, Sox2, Nestin, PDGFR, NG2 or any other related markers that will explain the nature of these clones. The origin of GBM is shown to be of NSC or OPC lineage, and GBM is known to harness cancer stem cells within. Also in the discussion section (line 275-278) the authors discuss and refer to the importance of DNMT in the transformation process. The authors also see a decrease in GFAP expression, which could be indicative that these cells are more stem cell like. Therefore it is vital to address the stem cell nature of these clones.

6.       Minor correction: I suggest the authors get professional service or have a native English writer evaluate their manuscript. There are various minor improvements that can be incorporated. Few examples are (not all):

line 373, recorded instead of “taken”

line 374: sensitivity instead of “sensibility”

line 343, 352, 368: please add colons after the first sentence

Author Response

"Please see attachment." 

Reviewer 2 Report

Title

Cellular model of malignant transformation of primary human astrocytes induced by deadhesion/readhesion cycles.

Concise summary

The authors describe a cellular model of malignant transformation of human astrocytes induced by cycles induced by repeated sequential sequential cycles of forced anchorage procedure.

Major Comments and Questions

1.K188 human astrocytes are human brain progenitor-derived astrocytes. K188 cells that were resistant to apoptosis induced by repeated adhesion blockage (anoikis resistance) showed new properties as spheroid formation what correlates with this undifferentiated profile. These cells were cultured giving rise to the astrocyte sublines. Question: Although glioma stem cells show anoikis resistance, is to what extent the tumor astrocytes obtained through this stressed experimental procedure can be extrapolated to the clinical setting of glioblastoma?

2.The authors state that GFAP expression is lost during the deadhesion/readhesion cycles even after the first adhesion blockage cycle. The loss of this constitutional intermediate filament is a very relevant fact. It is certain that glioblastoma show reduction in the immunohistochemical expression of this protein in respect to normal astrocytes, but the total loss of this filament is unusual. Questions. It can be estimated that the experimental procedure could induce an “excessive transformation” of the progenitor cells to a non-astrocytic cell. To what extent can these transformed cells continue to be considered tumor astrocytes? Could astrocyte sublines be considered as an epithelial to mesenchymal transition?

3.The authors state that EGFR expression was observed decreased in astrocyte-derived cell lines in relation to the parental primary astrocytes. However, the EGFR overexpression (associated to EGFR amplification) is frequently observed as a characteristic of glioblastomas. Question: How the authors explain this contradictory EGFR expression between astrocyte sublines respect to the human glioblastoma?

4.An immunohistochemical characterization of the astrocyte transformed cells have not been carried out in the study. It is recommended to obtain an immunohistochemical characterization of the astrocyte sublines including E-cadherin, vimentin, GFAP in order to know the phenotype of this transformed cells and what extend the K188 transformed astrocytes can be extrapolated to glioblastoma.

5.The authors state that the expression of DNMT genes related to DNA methylation, which codes for DNA methyltransferase, is increased. However, a report suggests that other genes different to DNMT genes may be linked to polycomb-associated de novo methylation. (Etcheverry et al. DNA methylation in glioblastoma: impact on gene expression and clinical outcome. BMC Genomics 11, 701 (2010). Can the authors explain to what extent the DNA methylation induced by DNMT genes in astrocyte sublines can be extrapolated to human glioblastoma?

6.It is stated that the undifferentiated cell clones demonstrated high ability to proliferate and high rate of cell migration. However, glioblastoma cells show either proliferative or migratory abilities. In fact, it is postulated that tumor cells show a mesenchymal phenotype when they adopt an invasive profile. Question. Could the authors to explain if these apparently discrepant activities could be observed in the astrocyte cell-lines obtained through repeated adhesion blockage?

7.The authors conclude that in their “cell model of GBM transformation…”. It is known that the oncogenesis of glioblastoma is a complex process which implies brain stem cells and microenvironment. Question: Which data support that the established astrocyte sublines correspond to glioblastoma?

In conclusion, I consider that it is an interesting and original work that give relevant information about a possible effect of deadhesion cycles of astrocytic transformation. However, the results should be relativized because the are many variables that have not been conveniently studied in a relatively simple experimental model.

Author Response

"Please see attachment." 

Reviewer 3 Report

The article investigates the malignant transformation of astrocytes through multiple cycles of deadhesion and adhesion. In general, the topic is very interesting and could contribute to the understanding of tumor development of astrocytomas. The experimental design and scope also seem appropriate. However, in some experiments the controls are missing and the results are incomplete analyzed, which would be necessary to understand the findings. The following points should be added or improved:

Major points:

Introduction

Line 50 - high metastasis is wrong; metastases in GBMs are rare - recurrences and high invasiveness are more likely meant.

The aim of study should be clearly stated at the end of the introduction.

Results

In general, the results are poorly and inaccurately described. Quantification to classify the data is missing for all results.

In Figure 1 A, it looks like the confluence of cells increases from astrocytes to Ast4c. How can this be explained?

In Figure 2 and 3 and in sections 2.2. and 2.3. respectively, the controls are missing. Why was the comparison with astrocytes and Ast1c-Ast4 not shown here?

The magnifications of microscope images and the figure legend of Figure 2 is confusing. The magnification of A and B is unclear. The authors mentioned in Figure 1 a magnification of 40x and now in Figure 2C a magnification of 4x. Please check this and improve figure legends. Scale bars should be inserted into all microscope images.

Are astrocytes and AST1c-AST4c cells able to form colonies or spheroids under these conditions?

How sensitive are these cells to TMZ treatment compared to clones (ASTcl3, ASTcl6, ASTcl7)?

What are the IC50 values of primary astrocytes and AST1c-AST4c cells?

What means “Bars represent the means ± standard deviations of the fold change relative to the control at time zero”? Please define the control.

In Figure 6, the white fields in the heat map are not defined. The significance of the data is unclear.

Discussion

Classification in the model -initiation, promotion and progression- mentioned in the introduction is missing in the discussion.

An outlook is missing.

Only the results of Figure 6 are discussed.

Materials and Methods

The size of PCR products of CCND2, Actin and HPRT should be checked. The sequence of CCND2 forward primer is identical the reverse primer? This raises questions about the correctness of the expression analyses.

Minor points:

The affiliations do not include the assignment to the corresponding author.

Line 18 - space is missing before “Here”

Line 88 - ASTc e - surely an "and" should be there?

Line 95 – “locus” is written in a different font

Line 132 – Please change “SFB” to “FBS”

Line 155 and 157 – “48h” space is missing

Line 179 - I have never encountered the spelling (conc) like that - is that how it is done?

Line 189 / 190 - abbreviations should be explained

Line 216 – Please remove “)“

Line 233 - Please remove comma between citations

Line 259 - Please remove dot after citations

Line 275 - expression written in italics

Line 285 - EGFR now explained here, more appropriate would be line 189

Line 424 - abbreviations not explained

Author Response

"Please see attachment." 

Round 2

Reviewer 2 Report

I think this article is an interesting and comprehensive research that gives relevant information about a possible effect of deadhesion cycles of astrocytic transformation. The results have been analyzed in a correct manner. The reviewer's comments have been answered correctly. The authors have corrected the text according to the reviewer suggestions. I think it is nice research, and the study has been developed as an adequate scientific approach. 

Reviewer 3 Report

I recommend the manuscript for acceptance and have no further comments.